# Inducible lncRNA transgenic mice reveal continual role of *HOTAIR* in promoting breast cancer metastasis

Qing Ma[1]*[†], Liuyi Yang[1†], Karen Tolentino[2†], Guiping Wang[2], Yang Zhao[2], Ulrike M Litzenburger[2], Quanming Shi[2], Lin Zhu[1], Chen Yang[3], Huiyuan Jiao[3], Feng Zhang[3], Rui Li[2], Miao-Chih Tsai[2], Jun-An Chen[4], Ian Lai[5,6], Hong Zeng[5,6], Lingjie Li[3]*, Howard Y Chang[2,7]*

[1]Shenzhen Key Laboratory of Synthetic Genomics, Guangdong Provincial Key Laboratory of Synthetic Genomics, CAS Key Laboratory of Quantitative Engineering Biology, Shenzhen Institute of Synthetic Biology, Shenzhen Institutes of Advanced Technology, Chinese Academy of Sciences, Shenzhen, China; [2]Center for Personal Dynamic Regulomes and Program in Epithelial Biology, Stanford University School of Medicine, Stanford, United States; [3]Department of Histoembryology, Genetics and Developmental Biology, Shanghai Key Laboratory of Reproductive Medicine, Key Laboratory of Cell Differentiation and Apoptosis of Chinese Ministry of Education , Shanghai Jiao Tong University School of Medicine, Shanghai, China; [4]Institute of Molecular Biology, Academia Sinica, Taipei, Taiwan; [5]Transgenic, Knockout, and Tumor Model Center, Stanford University School of Medicine, Stanford, United States; [6]Stanford Cancer Institute, Stanford University School of Medicine, Stanford, United States; [7]Howard Hughes Medical Institute, Stanford University, Stanford, United States

*For correspondence:
qing.ma@siat.ac.cn (QM);
lingjie@shsmu.edu.cn (LL);
howchang@stanford.edu (HYC)

[†]These authors contributed equally to this work

**Abstract** *HOTAIR* is a 2.2-kb long noncoding RNA (lncRNA) whose dysregulation has been linked to oncogenesis, defects in pattern formation during early development, and irregularities during the process of epithelial-to-mesenchymal transition (EMT). However, the oncogenic transformation determined by *HOTAIR* in vivo and its impact on chromatin dynamics are incompletely understood. Here, we generate a transgenic mouse model with doxycycline-inducible expression of human *HOTAIR* in the context of the MMTV-PyMT breast cancer-prone background to systematically interrogate the cellular mechanisms by which human *HOTAIR* lncRNA acts to promote breast cancer progression. We show that sustained high levels of *HOTAIR* over time increased breast metastatic capacity and invasiveness in breast cancer cells, promoting migration and subsequent metastasis to the lung. Subsequent withdrawal of *HOTAIR* overexpression reverted the metastatic phenotype, indicating oncogenic lncRNA addiction. Furthermore, *HOTAIR* overexpression altered both the cellular transcriptome and chromatin accessibility landscape of multiple metastasis-associated genes and promoted EMT. These alterations are abrogated within several cell cycles after *HOTAIR* expression is reverted to basal levels, indicating an erasable lncRNA-associated epigenetic memory. These results suggest that a continual role for *HOTAIR* in programming a metastatic gene regulatory program. Targeting *HOTAIR* lncRNA may potentially serve as a therapeutic strategy to ameliorate breast cancer progression.

## Editor's evaluation

A valuable new mouse model was developed for studying the functional effects of overexpressing *HOTAIR* and the mechanism of action of *HOTAIR*, and used to demonstrate overexpression of *HOTAIR* promoted breast cancer metastasis to the lung. Mechanistically, *HOTAIR* overexpression changed the chromatin from a closed, inactive form to an open, active form, activating specific genes and pathways and causing an altered cellular state that allowed the cells to metastasize; Importantly, the breast cancer cells depended on continuous *HOTAIR* expression, as *HOTAIR* does not trigger an epigenetic memory upon transient induction. The study offers fundamental insights, based on compelling data.

## Introduction

Long noncoding RNAs (lncRNAs) are extensively transcribed from mammalian genomes, and may actively participate in diverse biological processes through means beyond protein production (*Rinn and Chang, 2012*). Tens of thousands of lncRNAs have been identified by high-throughput RNA sequencing, but only a small percentage of these species have been functionally characterized (*Cabili et al., 2011*; *Forrest et al., 2014*; *Djebali et al., 2012*; *Iyer et al., 2015*; *Vancura et al., 2021*; *Zhou et al., 2021*). Once thought to be byproducts of mRNA splicing with no known biological activity within the cell, a handful of lncRNAs have been demonstrated to act in regulating gene expression as well the deposition of epigenetic modifications (*Kashi et al., 2016*). Several lncRNAs serve important roles in gene regulation, including modulating gene transcription; controlling nuclear architecture; mRNA stability, translation, and deposition of post-translational modifications (*Yao et al., 2019*). LncRNAs perform these functions through a variety of mechanisms, such as by acting as molecular scaffolds to bring multiple proteins into proximity in three-dimensional space; as 'guides' like *Xist* and *HOTAIR* which recruit chromatin-modifying enzymes to the genome; and as bridges which link promoters to distal enhancers to alter gene expression levels (*Gupta et al., 2010*; *Lee, 2009*; *Rinn et al., 2007*; *Tsai et al., 2010*). Some lncRNAs, such as *LUNAR1* and *CCAT1*, facilitate or inhibit long-range chromatin interactions (*Ma et al., 2015*; *Trimarchi et al., 2014*; *Xiang et al., 2014*). The disruption of lncRNAs has been linked to the pathogenesis of human developmental defects, neuronal disorders, and cancer processes (*Fatica and Bozzoni, 2014*; *Huarte, 2015*; *Statello et al., 2021*; *Yang et al., 2021*).

Human HOX antisense intergenic RNA (*HOTAIR*) is a 2.2-kb RNA transcribed from the *HOXC* locus, and is expressed in posterior and distal body sites in accordance with its location within the *HOX* locus. *HOTAIR* binds both Polycomb Repressive Complex 2 (PRC2) and LSD1 complexes and recruits them to multiple genomic sites to promote coordinated H3K27 methylation and H3K4 demethylation, respectively, for gene silencing (*Chu et al., 2011*; *Jarroux et al., 2021*; *Rinn et al., 2007*; *Tsai et al., 2010*). Although the primary sequences of *HOTAIR* are poorly conserved between humans and mice, human and murine *HOTAIR* orthologs demonstrate some functional similarities and conserved RNA secondary structures (*Li et al., 2013*; *Somarowthu et al., 2015*). Nonetheless, the sequence divergence creates a challenge to study human *HOTAIR* in vivo.

The overexpression of *HOTAIR* in several types of human cancers, including breast cancers has been linked to poor survival and increased metastasis (*Gupta et al., 2010*; *Kim et al., 2013*; *Kogo et al., 2011*). Within breast cancer, *HOTAIR* has been found to be several hundred-fold more highly expressed in metastatic breast tumors and primary breast tumors destined to metastasize (*Gupta et al., 2010*). *HOTAIR* overexpression in primary breast cancers is a significant and independent predictor of subsequent metastasis and death across multiple independent cohorts (*Gupta et al., 2010*; *Sørensen et al., 2013*). A variety of evidence implicates *HOTAIR* as a key epigenetic regulator of breast cancer metastasis (*Gupta et al., 2010*; *Zhou et al., 2021*), but the details of the regulatory mechanism are unclear. Moreover, the mechanism of 'epigenetic memory' renders it possible for epigenetic regulators to induce a stable transcriptional program that is retained and propagated by cells without constitutive expression of the epigenetic modulators (*Amabile et al., 2016*; *Bintu et al., 2016*; *Nuñez et al., 2021*; *Park et al., 2019*; *Tarjan et al., 2019*; *Van et al., 2021*). For example, the histone methyltransferase Ezh2, a component of the PRC2 complex, can induce long-term silencing by creating a heterochromatin environment at endogenous human gene loci (*Holoch et al., 2021*; *O'Geen et al., 2019*). However, whether *HOTAIR*-mediated epigenetic and transcriptomic alterations participate in long-term epigenetic memory has not yet been demonstrated. Consequently, it

is unclear whether epigenetic changes mediated by *HOTAIR* can be recovered as a possible avenue of clinical treatment.

In this study, we generate a transgenic murine model with inducible human *HOTAIR* expression to mechanistically investigate how *HOTAIR* acts to promote breast cancer progression, and whether depletion of *HOTAIR* can serve as a therapeutic strategy for treating breast cancer in patients. Using the well-characterized MMTV-PyMT system for endogenous breast cancer development, we confirmed that the induced overexpression of *HOTAIR* facilitates breast cancer metastasis to the lung and an ongoing dependency on *HOTAIR* for this phenotype. We also found alterations in both the transcriptome and chromatin accessibility of multiple metastasis-associated genes after *HOTAIR* over-expression and such trend is partially reverted upon *HOTAIR* silencing, suggesting a continual role of *HOTAIR* in regulating gene expression by modifying chromatin accessibility.

## Results

### Generation of transgenic mouse with inducible *HOTAIR*

To better understand the physiological functions of *HOTAIR* in vivo, we generated a transgenic murine model in which human *HOTAIR* can be conditionally overexpressed upon Doxycycline (Dox) adminis-tration. Human *HOTAIR* cDNA was integrated downstream of a tetracycline response element (TRE) at the *HPRT* locus through inducible cassette exchange recombination in A2Lox.cre mouse embryonic stem cells (mESCs) (*Iacovino et al., 2011*). In this binary control system, the transcription of reverse tetracycline transactivator (rtTA) protein is stimulated by the addition of 1–2 µg/ml of Dox effector, leading to *HOTAIR* overexpression upon rtTA binding of TRE (*Figure 1A*). We confirmed that *HOTAIR* can be effectively induced in mESC with the addition of the drug (*Figure 1—figure supplement 1A*). In addition, we validated that upon withdrawing Dox treatment in these ES cells, *HOTAIR* overexpres-sion returned to baseline levels within 48 hr (*Figure 1B*). Based on this ES cell line, we then generated a *HOTAIR*-inducible expression mouse model (hereafter named HOTAIR-rtTA mouse) via ES micro-injection and subsequent chimera selection, backcrossing to confirm stable germline transmission of the desired construct. HOTAIR-rtTA mice fed with Dox consistently expressed exponentially higher levels of *HOTAIR* compared to untreated control mice, as verified through quantitative reverse tran-scription PCR(qRT-PCR) (*Figure 1C*).

In order to investigate whether *HOTAIR* overexpression is associated with a visible phenotype, mice were fed with Dox water to induce high *HOTAIR* levels for as long as 1 year. We then evaluated the comprehensive morphology and metabolic features of different tissues. In most cases, we observed little to no change compared to wild-type littermates. The subtlety of these phenotypes suggests that *HOTAIR* overexpression in mice has little effect in a normal genetic background (*Figure 1—figure supplement 1B*).

### Elevated *HOTAIR* promotes breast cancer metastasis to the lung

High levels of *HOTAIR* are associated with poor survival and more aggressive breast cancers in xeno-graft models (*Gupta et al., 2010*). Human breast cancer cells which highly overexpress *HOTAIR* also display more metastatic and invasive properties compared to control cells (*Gupta et al., 2010*). To further study the function and mechanism of *HOTAIR* in promoting breast cancer progression, we engineered the inducible *HOTAIR* (iHOT) genetic system into a breast cancer model background. We crossed HOTAIR-rtTA mice to MMTV-PyMT (mouse mammary tumor virus-polyoma middle tumor-antigen) mice, a widely utilized breast cancer model in which mice spontaneously develop breast tumors at 8–12 weeks (*Figure 2A*, hereafter referred to as iHOT-PyMT mouse). This system allowed us to manipulate *HOTAIR* overexpression using Dox in mice with breast tumors and study *HOTAIR* function in tumor progression and metastasis in a controlled manner. We confirmed *HOTAIR* induction in this breast cancer model after Dox treatment using qRT-PCR, and found that *HOTAIR* transcript expression in PyMT mice containing the iHOT construct (iHOT⁺ Dox⁺ mice) was several hundred-fold higher compared to untreated iHOT mice (iHOT⁺ Dox⁻ mice) or control mice which did not contain the iHOT construct (iHOT⁻ mice) (*Figure 2B*). We continued to treat the mice with Dox for 3–4 months, exposing mice to continual *HOTAIR* overexpression, to investigate whether *HOTAIR* can affect breast tumor progression. First, we found that primary breast tumor weight and tumor numbers were not affected by continuous exposure to elevated levels of *HOTAIR* (*Figure 2C*). In contrast,

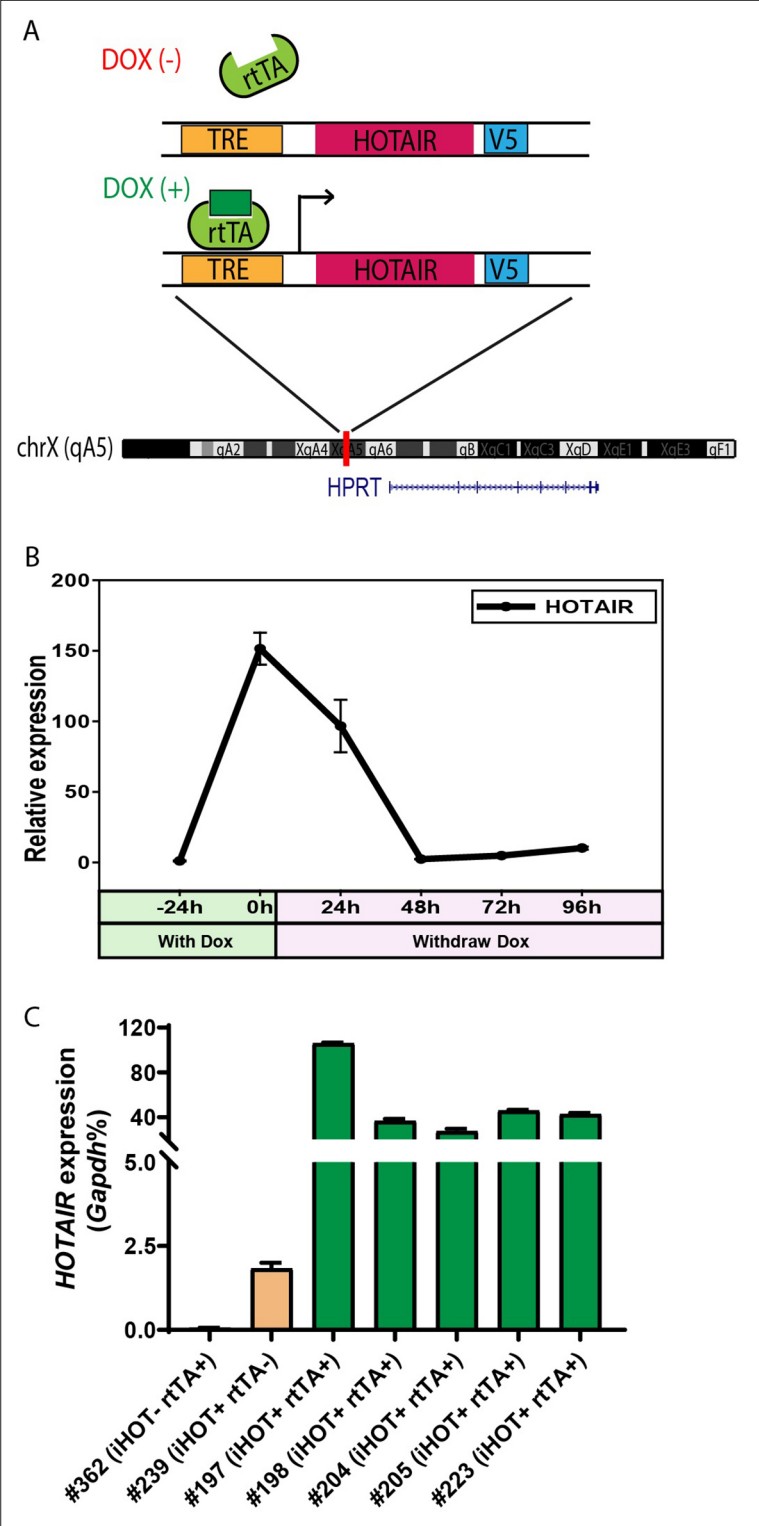

**Figure 1.** Generation of the inducible *HOTAIR* murine model. (**A**) Schematic description of the Doxycyclin (Dox)-inducible *HOTAIR* (iHOT) construct and insertion loci. (**B**) *HOTAIR* expression decreased to baseline levels after Dox withdrawal in inducible *HOTAIR* embryonic stem (ES) cells. Inducible *HOTAIR* ES cells were treated with 1 μg/ml Dox for 24 hr and then withdrew Dox treatment for 96 hr. Human *HOTAIR* expression was detected by qRT-PCR, normalized to *Gapdh* and showed as fold change relative to −24 hr. Values are means ± standard deviation (SD). (**C**) High levels of *HOTAIR* are induced in HOTAIR-rtTA mice (iHOT⁺ rtTA⁺) compare to the control group (iHOT⁻ rtTA⁻ or iHOT⁺ rtTA⁻) after Dox administration. All the mice were treated with Dox for about 10 days. RNA was

*Figure 1 continued on next page*

*Figure 1 continued*

extracted from mice tails. Human *HOTAIR* levels were quantified by qRT-PCR and calculated as percentages of the mouse *Gapdh* transcript. Values are means ± SD.

The online version of this article includes the following figure supplement(s) for figure 1:

**Figure supplement 1.** *HOTAIR* expression can be induced and inducible *HOTAIR* mice display no obvious phenotypes in different tissues.

---

after subsequent dissection of lung sections and hematoxylin and eosin staining (H&E staining), we observed a greater number of metastatic tumors and larger tumor volumes in the lungs of mice continually exposed to induced *HOTAIR* (iHOT⁺ mice + Dox treatment, *Figure 2D–E*, *Figure 3—figure supplement 1A*), indicating that *HOTAIR* overexpression drives metastatic progression. In addition, we also observed that Dox itself has no significant influence in tumor progression of MMTV-PyMT mice as previously reported (*Figure 2—figure supplement 1A*; *Nwokafor et al., 2016*; *Rumney et al., 2017*). To reduce the effect of endogenous mouse *Hotair*, we crossed our iHOT-PyMT mouse model to a murine *Hotair KO* mouse detailed in our previous work (*Li et al., 2013*). We then selected progeny carrying the appropriate genotypes. We found that expression of endogenous mouse *Hotair* was very low (*Figure 2—figure supplement 1B*) and that the knockout of this gene has no significant influence on breast tumor metastasis in MMTV-PyMT mice (*Figure 2—figure supplement 1C*). These results demonstrate that high *HOTAIR* expression promotes breast cancer metastasis in an autochthonous cancer model with intact immune system, consistent with previous clinical observations in patients and studies using xenograft models.

## Oncogene addiction to *HOTAIR* in breast cancer cells

As individual mice are variable in tumor progression, we isolated breast cancer cells from the primary tumors of iHOT-PyMT mice (abbreviated as iHOT⁺ cells) in order to gain further mechanistic insight into the functionality of *HOTAIR* (*Figure 3A*). In this manner, we can manipulate *HOTAIR* expression in vitro in late stage breast tumor cells in the same genetic background as in our iHOT-PyMT mouse model. We are also able to ascribe the potential impact of *HOTAIR* as intrinsic to cancer cells. *HOTAIR* RNA was hundreds-fold higher expressed in iHOT⁺ cells after Dox treatment and levels quickly decreased within 24 hr after Dox withdrawal, consistent with our observations in the in vivo HOTAIR-rtTA mouse (*Figures 1C and 3B*). In addition, we analyzed the subcellular localization of *HOTAIR* in iHOT⁺ cells using smFISH and found *HOTAIR* to localize to both the nucleus and cytoplasm. The signals intensity of *HOTAIR* increased upon Dox treatment (*Figure 3—figure supplement 1E*).

We next examined the effects of manipulating *HOTAIR* level in iHOT⁺ murine breast cancer cell lines in vitro. To assess the metastatic capacity of *HOTAIR* overexpressing cells as compared to control cells, we induced *HOTAIR* overexpression in iHOT⁺ breast cancer cells by Dox treatment for 7–18 days and subsequently performed a Matrigel invasion assay, which measures the ability of cells to migrate through a basement membrane like extracellular matrix (*Figure 3C*). We found that *HOTAIR* overexpression promotes the invasive capacity of breast cancer cells. In order to test whether *HOTAIR* overexpression evokes epigenetic 'memory' resulting in a phenotype of increased invasiveness, we performed a complete Dox withdrawal for 7–8 weeks. We observed that cellular invasion decreased after *HOTAIR* overexpressing cells underwent complete Dox withdrawal, suggesting that ongoing *HOTAIR* is required to promote increased cancer cell metastasis (*Figure 3C*). We also measured the growth curve of iHOT⁺ cells under these same three conditions and found that *HOTAIR* expression levels do not affect proliferation of iHOT⁺ breast cancer cells (*Figure 3—figure supplement 1C*).

To study the impact of *HOTAIR* overexpression on the metastatic potential of iHOT⁺ cells in vivo, we labeled iHOT⁺ cells with the firefly luciferase reporter gene *luc2*, enabling sensitive bioluminescence imaging of *HOTAIR* expression in live animals. We injected *luc2*-labeled iHOT⁺ cells exposed to Dox for 18 days, iHOT⁺ cells treated with Dox for 10 days and subsequently withdrawn from Dox, and untreated iHOT⁺ control cells through the tail vein of female SCID Beige mice. iHOT⁺ overexpression was sustained in live animals by feeding the experimental group Dox water over time. We then imaged live animals for luciferase bioluminescence every 2–3 days for around 2 weeks and measured the rates of lung colonization by injected iHOT⁺ tumor cells. As expected, mice injected with Dox-treated iHOT⁺ cells displayed significantly higher lung colonization after tail vein injection compared untreated control groups. Mice injected with cells that had been treated with Dox and

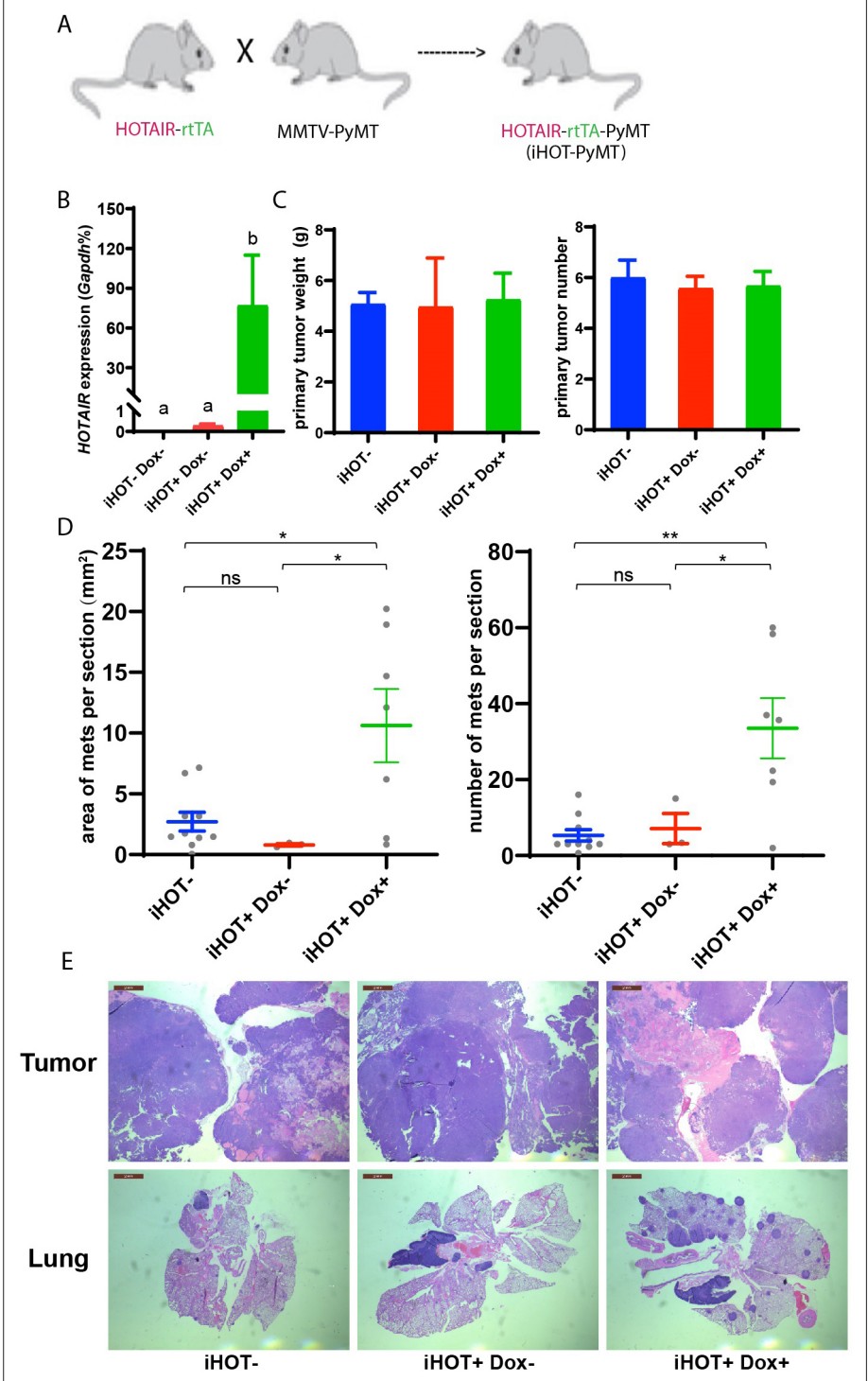

**Figure 2.** Induced *HOTAIR* promotes breast cancer metastasis in vivo. (**A**) Cross schema for generating the inducible *HOTAIR* construct in MMTV-PyMT genetic background. (**B**) qRT-PCR measurements demonstrate that Dox-treated iHOT-PyMT mice display significantly higher levels of *HOTAIR* expression compared to untreated control groups or controls lacking the iHOT construct. Values are means ± standard error of the mean (SEM), *n* = 2–3. One-way analysis of variance (ANOVA) was performed to compare the *HOTAIR* expression levels between each group (p < 0.05). (**C**) There are no statistically significant differences between primary breast tumor mass in grams or number of tumors between Dox⁺-treated *HOTAIR* overexpressing mice compared to the untreated control group or controls lacking the inducible *HOTAIR* construct. Values are means ± SEM, *n* = 7–9, mice are 3–4 months old. One-way ANOVA was performed between each group and showed no significant differences. (**D, E**)

*Figure 2 continued on next page*

*Figure 2 continued*

iHOT⁺ mice treated with Dox display an increased number of lung metastases with greater volume. Quantification of lung metastases based on hematoxylin and eosin (H&E) staining of lung and primary tumor sections of iHOT⁺ Dox-treated mice, untreated controls, and controls lacking the iHOT construct. Tumor area in sections was calculated by ImageJ. Dashes represent means ± SEM, *n* = 3–10. Spots represent every single value. One-way ANOVA was performed (*p < 0.05, **p < 0.01, ns: not significant). Scale bar = 2 mm.

The online version of this article includes the following figure supplement(s) for figure 2:

**Figure supplement 1.** Dox treatment or endogenous mouse *Hotair*, expressed at a very low level, have no significant effects on breast tumor metastasis.

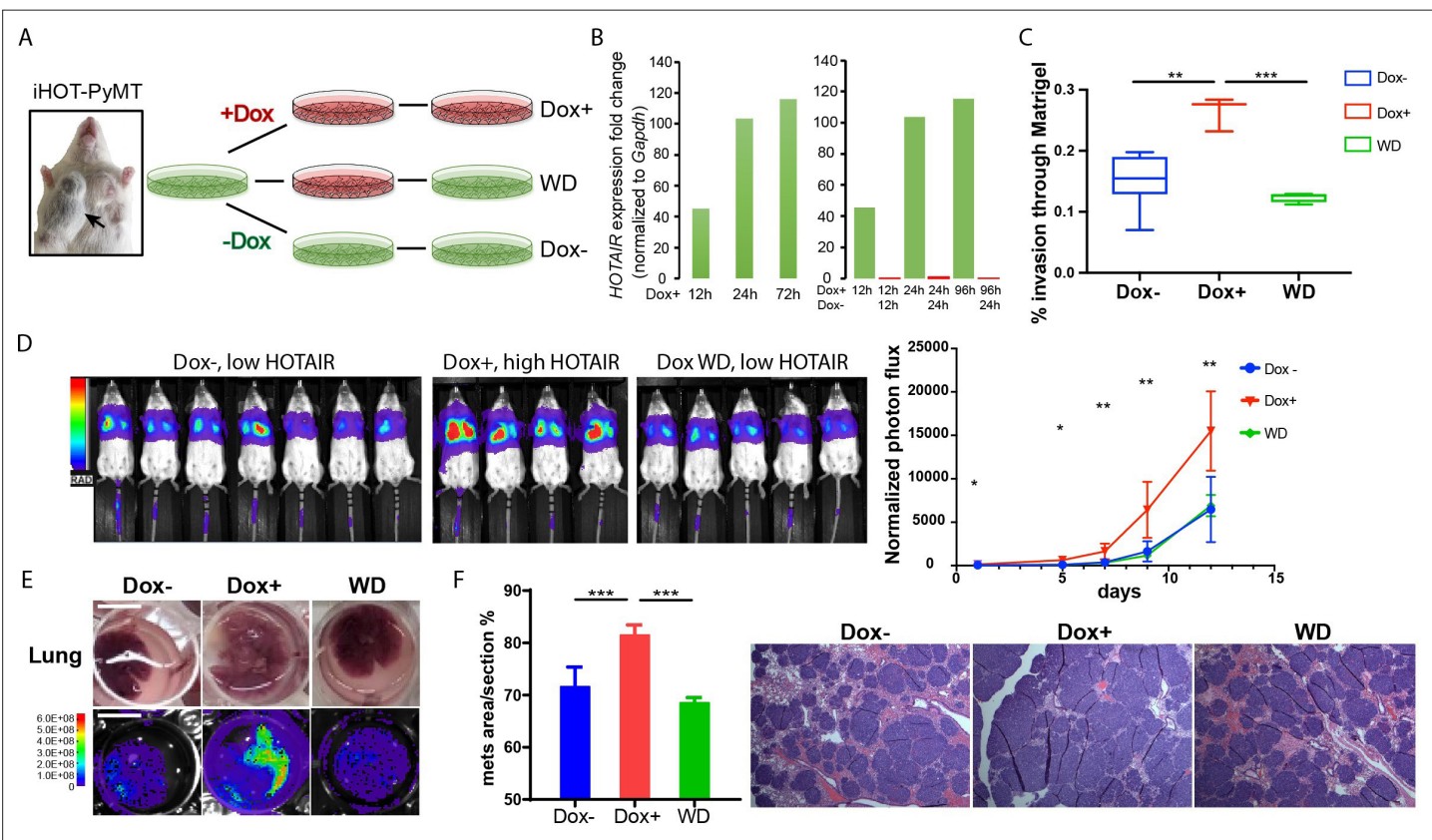

**Figure 3.** iHOT breast cancer cells display *HOTAIR* oncogene addiction. (**A**) Treatment strategy of iHOT⁺ cells to assess the physiological properties of *HOTAIR* overexpressing cells in vitro. iHOT⁺ cells are isolated from primary mammary tumors of iHOT-PyMT mice. There are three treatment groups: Dox⁻, Dox⁺, Dox withdraw after Dox treatment (WD). (**B**) qRT-PCR measurement of *HOTAIR* expression in iHOT⁺ cells after Dox treatment for indicated time and withdrawal at 12 or 24 hr. *HOTAIR* is effectively induced after Dox treatment and rapidly decreased after Dox withdrawal in iHOT⁺ breast cancer cells. (**C**) Dox-induced *HOTAIR* overexpression promotes increased invasive capacity in iHOT⁺ cells and the increased invasive capacity of these cells can be rescued when *HOTAIR* is restored to baseline levels after Dox withdrawal. Matrix invasion assay of iHOT⁺ breast cancer cells in Dox⁻, Dox⁺, Dox withdraw conditions was performed. Invasive capacity was measured by quantifying the displacement of iHOT⁺ breast cancer cells through Matrigel in media which did or did not contain 2 µg/ml Dox. Values were showed as box plot (*n* = 3–8). Student's *t*-test was performed between each group, **p < 0.01, ***p < 0.001. (**D**) Tail vein injection assay performed in SCID mice using iHOT⁺ breast cancer cells in Dox⁻, Dox⁺, Dox withdrawal conditions carrying a luciferase reporter. Dox-treated iHOT⁺ cells overexpressing *HOTAIR* display higher rates of lung colonization compare to the Dox⁻ group (*n* = 4–7, another batch in **Figure 3—figure supplement 1D**). Student's *t*-test was performed between Dox⁺ and Dox⁻ group at each time point, *p < 0.05, **p < 0.05. We show that the potential for lung colonization by iHOT⁺ cells decreases after Dox withdrawal. (**E**) Representative photos and bioluminescent imaging of lungs dissected from SCID mice at the 2-week time point after tail vein injection of iHOT⁺ cells. Scale bar = 1 cm. (**F**) Quantified percentage of metastatic lung tumor area in Dox⁻, Dox⁺, Dox withdrawal cells after tail vein injection in lung sections with HE staining. Values are means ± standard deviation (SD), *n* = 4–7. One-way analysis of variance (ANOVA) was performed between each group, ***p < 0.001.

The online version of this article includes the following figure supplement(s) for figure 3:

**Figure supplement 1.** Induced *HOTAIR* promotes tumor cell metastasis but has no significant effect on cell proliferation.

subsequently withdrawn displayed low lung colonization of fluorescent cells, similar to the biolu-minescence profile of mice injected untreated iHOT+ control cells (*Figure 3D, E*, *Figure 3—figure supplement 1D*). Altogether, our results suggest that the increased metastatic potential of *HOTAIR* overexpressing breast cancer cells requires ongoing *HOTAIR* activity and is abrogated by silencing *HOTAIR*.

We sacrificed injected mice at the 2-week time point and dissected lung tissue from the mice. We performed histologic analysis of lung sections using hematoxylin and eosin staining to visualize tumor metastasis. The lung sections of mice injected with *HOTAIR* overexpressing cells displayed a higher metastatic tumor burden compared to mice injected with *HOTAIR* withdrawn cells or untreated control cells (*Figure 3E, F*). In contrast, mice injected with *HOTAIR* withdrawn cells where *HOTAIR* expression level is returned to baseline display similar levels of lung colonization compared to injec-tion with untreated control cells. In brief, we conclude that breast cancer cells with elevated *HOTAIR* require persistent and sustained *HOTAIR* overexpression to retain a highly invasive and metastatic phenotype.

## *HOTAIR* is required for metastasis-associated chromatin reprogramming and gene expression

We hypothesize that *HOTAIR* dysregulation during breast cancer metastasis leads to an altered chro-matin landscape and subsequent changes in gene expression in affected cells by recruiting chromatin remodeling machinery such as the Polycomb complex (PRC2). PRC2 promotes chromatin compaction through catalyzing the trimethylation of histone H3 at lysine 27 (H3K27me3), a repressive histone mark. To this end, we analyzed gene expression patterns in iHOT+ cells treated with three conditions: Dox+ *HOTAIR* overexpressing cells which were consistently treated with Dox for several days; Dox$^{WD}$ cells which had been temporarily treated with Dox and later withdrawn from *HOTAIR* overexpression; and untreated control cells in which *HOTAIR* remained at baseline levels. We performed RNA-seq to gain insight into the molecular basis of the observed phenotypes within these experimental condi-tions. First, we checked *HOTAIR* expression levels in the RNA-seq data, which revealed that *HOTAIR* levels were elevated in the Dox+ group and reverted to basal level in the Dox$^{WD}$ group (*Figure 3—figure supplement 1B*). RNA-seq analysis revealed that there were 219 upregulated genes and 280 downregulated genes in the Dox-treated *HOTAIR* overexpressing group compared to the untreated control group (FDR <0.05, fold change >2, *Figure 4A*).

We then applied a gene ontology (GO) term analysis to these groups of differentially expressed genes (DEGs) and found terms consistent with invasion and metastasis phenotypes (*Figure 4—figure supplement 1A*). We discovered that GO terms such as 'epithelial cell proliferation' and 'Wnt Signaling Pathway and Pluripotency' were enriched in the upregulated gene group, and terms such as 'regulation of cell adhesion' were enriched in the downregulated gene group. Importantly, in the Dox withdrawal condition, we observed that most of these DEGs to revert to expression levels similar to that of the untreated control group. This result suggests that genes that are differentially expressed due to high levels of *HOTAIR* transcript in the system can be subsequently reversed to their orig-inal expression level upon *HOTAIR* withdrawal (*Figure 4A*). We visualized gene expression changes between experimental groups using a hierarchical clustering heatmap, revealing that the transcrip-tome patterns of Dox withdrawal cells are similar to Dox− cells (*Figure 4A*). We also compared DEGs within Dox$^{WD}$ vs. Dox− groups and found very few DEGs (2 upregulated and 15 downregulated). Furthermore, we observed little overlap between these DEGs compared to DEGs between Dox+ and Dox− groups (*Figure 4—figure supplement 1C, D*). This supports our previous in vivo observations, which imply that the increased invasiveness and metastatic properties of breast cancer cells can be reversed with *HOTAIR* withdrawal.

Since *HOTAIR* is known to reprogram chromatin status and alter chromatin modifications to promote cancer metastasis (*Gupta et al., 2010*), we performed an Assay for Transposase-Accessible Chromatin using sequencing (ATAC-seq) to assess genome-wide changes in the chromatin landscape induced by *HOTAIR* overexpression. High levels of *HOTAIR* induced by Dox treatment led to altered chromatin accessibility (*Figure 4B*). There are in total 1933 peaks with differential accessibility between all *HOTAIR* high Dox+ and all *HOTAIR* low Dox− cells, of which 1048 peaks are upregulated and 885 peaks are downregulated. We observed both up- and downregulated peaks to be mainly located in distal intergenic regions (about 50%). We found that a higher percentage of downregulated peaks

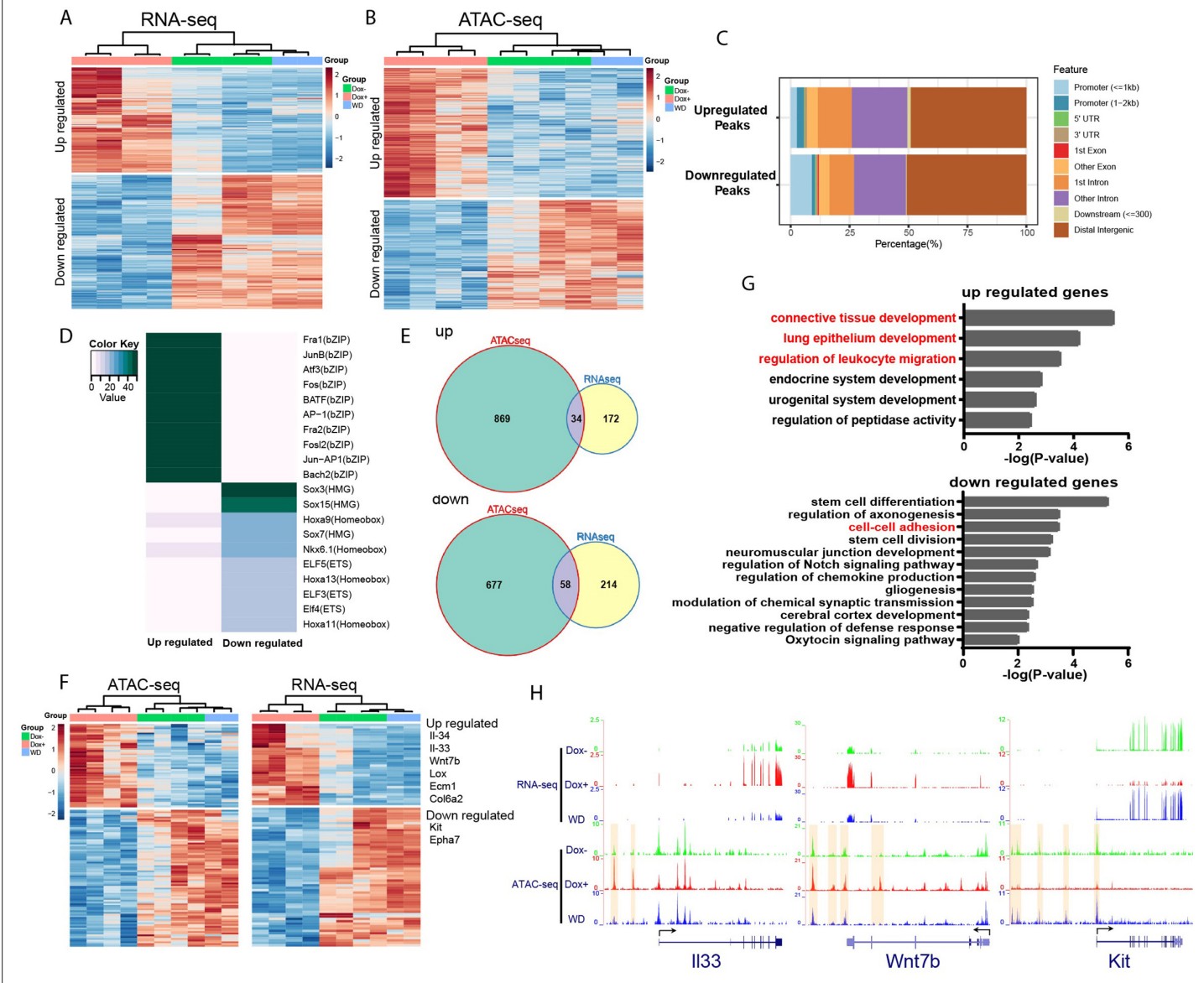

**Figure 4.** *HOTAIR* is required for metastasis-associated chromatin accessibility and genes expression. (**A**) Hierarchical clustering heatmap of RNA-seq data of iHOT+ cells subject to either Dox−, Dox+, or Dox withdrawal treatment. Dox-induced *HOTAIR* overexpression results in altered gene expression profiles. There are 499 genes differentially expressed between Dox+ and Dox− cells (FDR <0.05, fold change >2). Transcriptome patterns of Dox withdrawal cells are similar to Dox− cells. (**B**) Hierarchical clustering heatmap of differential ATAC-seq peaks of iHOT+ cells with Dox−, Dox+, Dox withdraw treatments. Dox-induced *HOTAIR* overexpression alters chromatin accessibility. There are 1933 differential peaks between Dox+ and Dox− cells. Chromatin accessibility landscape of differential peaks returns to Dox− status after Dox withdrawal. (**C**) Feature distribution of differentially regulated peaks. We observe significantly more downregulated peaks located in promoter regions (less than 1 kb to transcription start sites) compared to upregulated peaks (p < 0.01). (**D**) Enriched motifs of differential ATAC-seq peaks by HOMER motif analysis. Color coding indicates −log10 p values. (**E**) Venn diagram representing overlap between ATAC-seq differential peak-associated genes and differentially expressed genes by RNA-seq (up: upregulated genes; down: downregulated genes). (**F**) Heatmap of transcriptome and corresponding chromatin status of overlapping genes in (E) in Dox−, Dox+, and Dox withdrawal conditions. Listed on the right are several representative genes related to cancer cell metastasis. (**G**) Gene ontology of overlapping genes in (**E**). Terms highlighted in red are examples of terms related to cancer cell metastasis. (**H**) *HOTAIR* regulates chromatin accessibility and gene expression. Normalized ATAC-seq and RNA-seq sequencing tracks of metastasis-related genes *Il33*, *Wnt7b*, and *Kit*. Alteration of chromatin accessibility correlates well with transcriptional changes.

The online version of this article includes the following figure supplement(s) for figure 4:

**Figure supplement 1.** Gene ontology (GO) enrichment analysis of differential genes of Dox− vs. Dox+ in RNA-seq and ATAC-seq data and differential analysis between DoxWD and Dox−.

**Figure supplement 2.** Potential targets of *HOTAIR* and model of *HOTAIR*'s function.

occur in promoter regions less than 1 kb, indicating that *HOTAIR* most likely functions suppressively near promoter regions to modulate the expression of downregulated genes (*Figure 4C*).

To reveal possible functional pathways which *HOTAIR* participates in within the cell, we performed motif enrichment analysis by HOMER on the up- and downregulated peaks which we identified through ATAC-seq (*Figure 4D*). The top motif enriched in upregulated peaks involved many transcription factors associated with tumor metastasis, such as AP-1 family members Fra1, JunB, Atf3, and Fos (*Belguise et al., 2005*; *Hyakusoku et al., 2016*; *Milde-Langosch et al., 2004*; *Yuan et al., 2013*). In the top motif enriched in downregulated peaks, Sox7 was reported to play inhibitory roles related to cellular proliferation, migration, and invasion in breast cancer (*Stovall et al., 2013*). Previous work in pancreatic ductal adenocarcinoma has demonstrated that Sox15 is a potential tumor suppressor of the Wnt/β-catenin signaling pathway (*Thu et al., 2014*). These observations support our primary hypothesis that high levels of *HOTAIR* transcript promote breast cancer metastasis and suggest that *HOTAIR* might have functional relevance to those identified factors. We then analyzed peaks associated with known genes and found that cell samples subjected to Dox withdrawal also clustered together with Dox⁻ untreated control samples in a hierarchical clustering heatmap (*Figure 4B*). In addition, we identified very few differential peaks (0 upregulated and 40 downregulated) when comparing Dox⁻ and Dox^WD groups. We observed little overlap between these differential peaks compared to differential peaks within Dox⁺ and Dox⁻ groups (*Figure 4—figure supplement 1E, F*). These results are consistent with transcriptomic patterns that we have observed in previous experiments. GO analysis of these peaks associated with known genes showed enrichment of many terms related to cell adhesion (*Figure 4—figure supplement 1B*). For example, the terms 'regulation of cell adhesion' and 'leukocyte migration' were enriched in upregulated peaks associated with genes, whereas terms such as 'cell–cell adhesion', 'cell–substrate adhesion', and 'negative regulation of locomotion' were enriched in downregulated peaks associated with genes (*Figure 4—figure supplement 1B*). Consistent with our previous results in our RNA-seq experiments, the chromatin accessibility level of most differential peaks identified with ATAC-seq can be subsequently rescued by *HOTAIR* withdrawal in the Dox^WD *HOTAIR* low group (*Figure 4B*).

To further investigate downstream pathway components regulated by *HOTAIR*, we compared differentially accessible peaks in the immediate vicinity of DEGs, and from the overlapping set identified 34 upregulated genes and 58 downregulated genes associated with peaks. Many of the selected genes were linked to metastasis, such as *Il33*, *Il34*, *Wnt7b*, *Lox*, *Ecm1*, *Col6a2* (upregulated), *Kit*, and *Epha7* (downregulated) (*Figure 4E, F*). GO term analysis revealed these upregulated genes are associated with pathways such as 'connective tissue development', 'lung epithelium development', and 'regulation of leukocyte migration', while downregulated genes are associated with 'cell–cell adhesion', all of which are metastasis-related cellular processes (*Figure 4G*). As shown in examples from *Figure 4H*, both *Il33* and *Wnt7b* were upregulated with *HOTAIR* induction (Dox⁺) and reverted to low expression levels after *HOTAIR* withdrawal (WD). Conversely, *Kit* was downregulated after *HOTAIR* induction and reverted to high expression levels after *HOTAIR* withdrawal.

Several lines of evidence in the literature support the interpretation that our model accurately captures molecular changes corresponding to cancer progression and the development of metastatic cellular phenotypes. For example, ATAC-seq signals in the regulation region of the DEGs we identified display a consistent change in the ligand (*Figure 4H*) IL-33 and its receptor ST2, which form the IL-33/ST2 signaling pathway. This pathway has previously been linked with tumor progression, cellular invasion, and metastasis (*Kim et al., 2015*; *Larsen et al., 2018*; *Zhang et al., 2019*). The WNT/β-catenin pathway plays a critical role in tumorigenesis, and Wnt7b was reported to function in mediating metastasis in breast cancer (*Xu et al., 2020*; *Yeo et al., 2014*). Kit, also known as c-Kit, CD117 (cluster of differentiation 117), or SCFR (mast/stem cell growth factor receptor), is one of the receptor tyrosine kinases (RTK), and has been previously reported to be involved in tumor migration or invasion (*Golubovskaya and Wu, 2016*). Altogether, these results suggest that *HOTAIR* transcript regulates the expression of multiple metastasis-associated genes by modifying their chromatin accessibility to promote tumor metastasis.

We observed that some adhesion-associated GO terms were enriched for *HOTAIR*-regulated genes and that the expression level of several EMT hallmark genes such as *Lox*, *Ecm1*, and *Col6a2* were significantly altered with *HOTAIR* overexpression (*Figure 4G, F*). To investigate the metastatic phenotype of iHOT⁺ cells, we performed immunofluorescence staining of prototypical EMT markers.

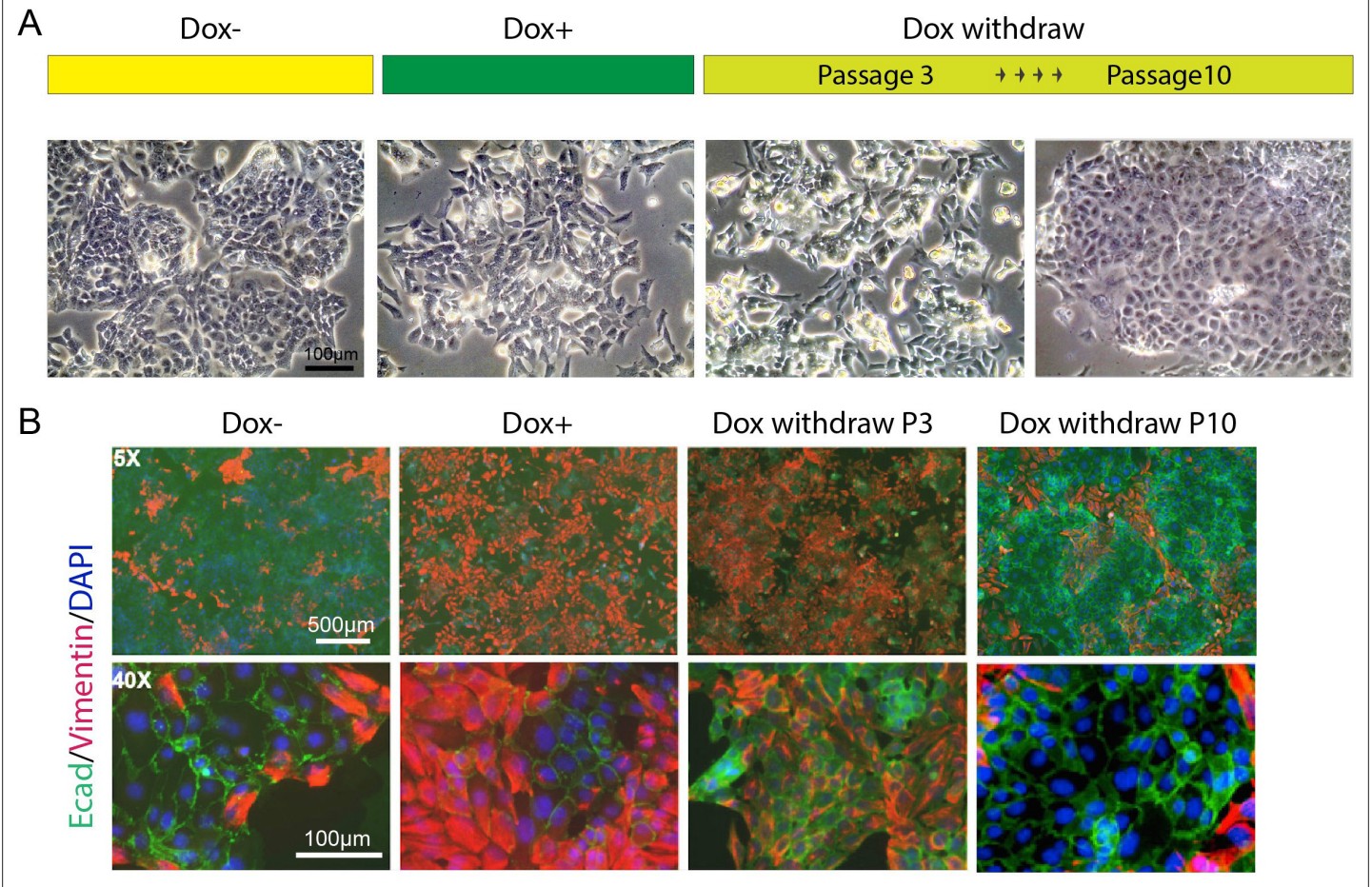

**Figure 5.** *HOTAIR* continually enforces epithelial–mesenchymal transition of breast cancer cells. (**A**) Scheme demonstrating how iHOT+ cells were treated with Dox and microphotographs of cell morphology in each condition. (**B**) Immunofluorescence staining of E-cadherin (green), Vimentin (red), and DAPI (blue) of iHOT+ cells in each condition.

After Dox treatment, the mesenchymal marker Vimentin was induced and the epithelial marker E-cadherin was reduced. In addition, the iHOT+ cells also displayed more spindle-like morphology and cell detachment from epithelioid colonies (*Figure 5A, B*). When Dox was withdrawn for a brief period over three cellular passages, Vimentin was still expressed albeit at a reduced level. However, when Dox was withdrawn for an extended period of time over 10 cellular passages, Vimentin was largely abrogated and E-cadherin re-expressed to similar levels as in untreated cells, and Dox withdrawn cells returned to epithelial morphology (*Figure 5A, B*). These results demonstrate that iHOT+ cells undergo EMT when exposed to *HOTAIR* overexpression as a result of treatment with dox, and recover their original epithelial states after longer periods of withdrawal from Dox, which is consistent with the invasion and metastasis phenotypes we previously observed. However, it appears that effects modulated by *HOTAIR* might persist for some time. For example, as *HOTAIR* expression is quickly restored to basal levels within 24 hr after Dox withdrawal (*Figure 3B*), elevated Vimentin expression endures over several days lasting as long as three passages (*Figure 5B*).

Collectively, our results demonstrate that the cellular function of *HOTAIR* as a regulator of gene expression is conserved between mammalian lineages and confirm that exposure to sustained high levels of *HOTAIR* over time facilitates increased EMT, invasiveness and metastatic capacity in breast cancer cells. We also observe alterations in both the cellular transcriptome and the chromatin accessibility landscape of multiple metastasis-associated genes after *HOTAIR* overexpression. Subsequently silencing *HOTAIR* expression for enough time abrogates this metastatic phenotype, as well as any resulting alterations in transcriptome and chromatin accessibility, indicating that these *HOTAIR*-modulated alterations have not yet formed persistent epigenetic memory. Thus, *HOTAIR* is required

in an ongoing manner to reprogram oncogene expression. Our results also suggest that targeting overexpressed *HOTAIR* transcript can potentially serve as a therapeutic strategy in breast cancer patients to limit metastatic progression.

## Discussion

Overexpression of the lncRNA *HOTAIR* is significantly correlated with tumor progression and poor prognosis in many kinds of tumors and is a powerful predictor of tumor metastasis. As such, targeting *HOTAIR* overexpression is of clinical interest as a potential therapy for cancer patients (*Gupta et al., 2010*; *Mozdarani et al., 2020*). Hence, there exists a critical need to systematically interrogate and understand the physiological functions and mechanisms of action of *HOTAIR* in model organisms. The HOTAIR-rtTA-PyMT transgenic mouse generated in this paper provides a model to analyze the functional role of human *HOTAIR* in vivo within the context of breast cancer metastasis.

Murine *HOTAIR* and human *HOTAIR* transcripts are conserved in structure rather than primary sequence (*He et al., 2011*; *Yu et al., 2012*). It was previously reported that human *HOTAIR* binds to both the PRC2 and LSD1 chromatin remodeling complexes, acting as a scaffold to target these complexes to the HOX loci and enforcing a silenced chromatin state at this position (*Tsai et al., 2010*). Human *HOTAIR* may have additional gene silencing mechanisms (*Meredith et al., 2016*). Murine *HOTAIR* has been observed to function as a trans-acting regulator similar to human *HOTAIR*, capable of acting in trans to influence the activities of distal genes to its site of origin (*Li et al., 2013*). However, there exists no direct evidence of this functional conservation. In our inducible human *HOTAIR* murine model, human *HOTAIR* transcript overexpression was found to promote breast cancer metastasis in vivo, consistent with previous observations in patients and human breast cancer cell lines. Based on our ATAC-seq data, we find significant changes in the chromatin landscape when human *HOTAIR* is upregulated in mouse tissue, indicating that human *HOTAIR* functions through a similar mechanism as murine *HOTAIR* by binding chromatin remodeling machinery and targeting the biochemical activities of these complexes to specific genomic loci. We postulate that this reprogramming of the underlying chromatin by *HOTAIR* leads to altered cellular function, which allows the cell to metastasize and thrive at ectopic sites.

Many lncRNAs function in regulating epigenetic modifications, some of which are also involved in the establishment of persistent epigenetic memory (*Arunkumar et al., 2022*; *Begolli et al., 2019*; *Fok et al., 2018*; *Wei et al., 2017*). For example, recent work showed that transcription of locus 8q24-derived oncogenic lncRNAs such as *PCAT2* could recruit centromeric protein-A (CENP-A), a variant of Histone H3, promote the ectopic localization of CENP-A, and alter epigenetic memory at a fragile chromosomal site in human cancer cells (*Arunkumar et al., 2022*). In the current study, we investigated whether *HOTAIR*-mediated epigenetic alterations participate in enacting long-term epigenetic memory using transgenic iHOT[+] tumor cells. Our results demonstrate that a complete Dox withdrawal could restore all alterations in transcriptomes, chromatin states or metastatic phenotypes. Our data imply that the presence of *HOTAIR* transcript is required to promote cancer metastasis and propagate changes in the chromatin landscape. This indicates that *HOTAIR* is continually needed to enforce cellular transcriptional programs—a 'lncRNA addiction'. In contrast, *Xist* lncRNA is involved in establishment of X chromosome inactivation in female cells but is largely dispensable for the maintenance of the inactive state over subsequent cell divisions and embryonic development (*Yu et al., 2021*). In this regard, the role of *HOTAIR* is more akin to a transcription factor that mediates a large-scale but reversible program of gene expression. We demonstrated that restoring *HOTAIR* expression level to baseline from a highly overexpressed state in iHOT[+] breast cancer cells isolated from the primary tumors of iHOT-PyMT mice could rescue expression of most altered genes and chromatin accessibility (*Figure 4A, B*), as well as restoring the overall physical phenotype of the organism (*Figures 3C–F and 5*). Our results suggest that persistent and sustained *HOTAIR* overexpression engender high metastatic potential within cancerous cells, further implying *HOTAIR*'s utility as a therapeutic target.

Our mouse model of human *HOTAIR* function provides an in vivo system to test small molecules or antisense oligonucleotides (ASOs) as potential therapeutics in the future. LncRNA can be targeted by multiple approaches: (1) siRNAs or ASOs can be used to silence genes in a sequence-specific manner to decrease lncRNA levels post-transcriptionally; (2) Modulation of lncRNA expression by the CRISPR/Cas9 System; (3) Inhibition of RNA–protein interactions or prevention secondary structure formation (*Arun et al., 2018*). Several studies demonstrated that knockdown of *HOTAIR* can be achieved using

siRNA or ASO strategies in vitro (*Lennox and Behlke, 2016*; *Rinn et al., 2007*; *Tsai et al., 2010*; *Yang et al., 2011*), but targeting lncRNAs in vivo continues to present significant challenges. A peptide nucleic acid (PNA)-based approach was reported to block the EZH2-binding domain of *HOTAIR*, inhibiting *HOTAIR*-EZH2 activity and subsequently decreasing invasion of ovarian and breast cancer cells and resensitizing resistant ovarian tumors to platinum-based chemotherapy (*Özeş et al., 2017*).

In addition, several drugs have been recognized to indirectly downregulate *HOTAIR* expression (*Mozdarani et al., 2020*). For example, the isoflavone-based drugs Calycosin and Genistein inhibit *HOTAIR* expression by repressing Akt pathway upstream of *HOTAIR* transcription (*Chen et al., 2015*; *Özeş et al., 2017*). Direct or indirect suppression of the WNT pathway was also reported to downregulate *HOTAIR* expression in cancer cells (*Carrion et al., 2014*; *Wang et al., 2015*).

Using a combination of RNA-seq and ATAC-seq assays, we identified a candidate gene set with altered expression or chromatin status in the context of *HOTAIR* overexpression, which were rescued when *HOTAIR* expression level was restored. This gene set encompasses the direct or indirect molecular targets of *HOTAIR*. We also utilized published *HOTAIR* ChIRP-seq data to analyze *HOTAIR*'s potential direct targets by interrogating whether *HOTAIR*-associated genes are altered with *HOTAIR* induction in iHOT cells. The previous study identified 832 *HOTAIR* occupancy sites genome-wide which are associated with 1345 genes (*HOTAIR* targets) in total (*Chu et al., 2011*). We compared *HOTAIR* targets established from previous literature with DEGs and differential peak-associated genes identified from our RNA-seq and ATAC-seq assays on Dox⁻ vs. Dox⁺ iHOT cells. We observed significant overlaps between *HOTAIR* targets and ATAC-seq differential peak-associated genes (207 genes, ~13% of ATAC-seq differential peak-associated genes, *Figure 4—figure supplement 2A*).

Among these shared genes, the chromatin accessibility of 116 genes were upregulated and 91 were downregulated. Our GO analysis uncovered terms such as cell adhesion, EMT, or migration (*Figure 4—figure supplement 2A*). We also noted significant overlap between *HOTAIR* targets and DEGs in our RNA-seq data (34 total, 13 genes were upregulated and 21 genes were downregulated in Dox⁺ cells compared to control, *Figure 4—figure supplement 2B*). GO analysis of these genes also revealed associations with multiple EMT and migration related terms (*Figure 4—figure supplement 2B*). Furthermore, we compared the *HOTAIR* targets to genes regulated by *HOTAIR* through chromatin accessibility, which were represented as overlapping genes shared between our ATAC-seq and RNA-seq datasets (*Figure 4E*). We identified 12 genes out of a total of 92 genes (~13%) which may be direct targets regulated by *HOTAIR* in iHOT cells (*Figure 4—figure supplement 2C*). Interestingly, almost all of these genes were associated with migration. Genes shown to promote migration such as *Ecm1*, *Gpr39*, and *Padi1* were upregulated and genes shown to suppress migration such as *Limch1* were downregulated (*Figure 4—figure supplement 2C*). These results suggest that *HOTAIR* regulates the expression of multiple cell adhesion- and EMT-associated genes by regulating their chromatin accessibility to promote tumor metastasis (*Figure 4—figure supplement 2D*).

Previous work showed that *HOTAIR* regulates genome chromatin landscape through influencing both H3K27me3 and H3K27ac modifications (*Gupta et al., 2010*; *Jarroux et al., 2021*; *Song et al., 2019*). Performing a ChIP-seq assay focused on H3K27me3 and H3K27ac combined with a ChIRP-seq assay targeting *HOTAIR* in iHOT⁺ cells would give insights into the direct *HOTAIR* targets. Future work to delineate the direct targets and cofactors of *HOTAIR* within these gene sets will provide further insight into the molecular mechanism and biological function of the *HOTAIR* transcript.

# Materials and methods

## Animals

### Generation of the inducible *HOTAIR* mESC line

To generate the inducible *HOTAIR* mESC line, we first cloned human *HOTAIR* cDNA into p2Lox plasmid (p2Lox-hHOTAIR), then transfected the plasmid into A2Lox.cre mESC lines. The stably integrated cell line was selected through the inducible cassette exchange method, described with details in *Iacovino et al., 2011*.

### Generation of HOTAIR-rtTA mouse

To generate the inducible *HOTAIR* transgenic mouse model, we performed micro-injection of the ES cells into the mouse blastocyst, which was allowed to develop into the chimeric animal. After

backcrossing to pure C57BL/6 mice and performing genotyping selection, we confirmed that the transgene was stably integrated and germline transmittable animals were picked-up and maintained for the subsequent study.

### Generation of iHOT-PyMT mouse

To generate a breast cancer animal model with conditional expression of human *HOTAIR*, we crossed the HOTAIR-rtTA line with a murine breast cancer mouse line MMTV-PyMT (FVB/N-Tg (MMTV-PyVT) 634Mul/J #2374, Jackson Laboratory). Through several rounds of crossing and genotyping, the triple-positive mice (i.e., *HOTAIR+*, *rtTA+*, and *MMTV+*) were selected for further study.

All mice were bred in the Stanford University Research Animal Facility in accordance with the guidelines. All mouse work was performed according to IACUC approved protocols at Stanford University (APLAC-14046).

Dox-treated mice were fed with doxycycline at the concentrations of 5 g/l supplemented with 50 g/l sucrose via drinking water.

## Tumor cell isolation and DOX treatments

iHOT$^+$ cells were derived from the primary tumors of iHOT-PyMT mice as described in the following steps. The primary tumors of iHOT-PyMT mice were dissected and chopped into small pieces. The tissues were then digested using 5-ml collagenase buffer (0.5% collagenase I in Hanks' Balanced Salt Solution (HBSS)) per gram for 1–1.5 hr at 37°C with moderate shaking (~200 rpm). The suspended cell mixture was then spun at 600 rcf for 2–10 min. The pellet of small epithelial sheet was resuspended in 0.25% Trypsin for 10 min and neutralized with 10% fetal bovine serum (FBS)/Dulbecco's Modified Eagle Medium (DMEM) and 1× DNase 10 min at 37°C. The dissociated cells were passed through a 0.45-μm cell strainer, and the cells were pelleted at 450 rcf for 5 min. The cells were then washed with phosphate-buffered saline (PBS) and pelleted again. The cells were then plated and cultured in DMEM/F-12 media with 5% Tet System Approved FBS, 1% P/S penicillin–streptomycin, 5 μg/ml insulin from bovine, and 1 μg/ml hydrocortisol. The genotype of the iHOT-PyMT mouse was confirmed by PCR. The inducible *HOTAIR* overexpression of the iHOT cells was also confirmed by qRT-PCR. We perform routine testing for mycoplasma contamination, which were negative.

Three conditions of iHOT$^+$ cells were used in this study: Dox$^+$, Dox$^-$, and Dox withdrawal (Dox$^{WD}$). In the Dox$^+$ group, iHOT$^+$ cells were treated with 2 μg/ml DOX for 7 or 18 days; in the Dox$^-$ group, iHOT$^+$ cells were treated with solvent control for 18 or 59 days; and in the Dox$^{WD}$ group, iHOT$^+$ cells were first treated with DOX for 10 days then removed for another 49 days.

## qRT-PCR

Total RNA from mouse tails or iHOT$^+$ cells was extracted using TRIzol and the RNeasy mini kit (Qiagen). RNA levels (starting with 50–100 ng per reaction) for a specific gene were measured using the Brilliant SYBR Green II qRT-PCR kit (Strategene) according to the manufacturer's instructions. All samples were normalized to mouse *Gapdh*.

Primers used:

> *hHOTAIR*-F: GGTAGAAAAAGCAACCACGAAGC
> *hHOTAIR*-R: ACATAAACCTCTGTCTGTGAGTGCC
> *mHotair*-F: CCTTATAAGCTCATCGGAGCA
> *mHotair*-R: CATTTCTGGGTGGTTCCTTT
> *mGapdh*-F: CTGGAGAAACCTGCCAAGTA
> *mGapdh*-R: TGTTGCTGTAGCCGTATTCA

## Invasion assay

The matrigel invasion assay was done using the Biocoat Matrigel Invasion Chamber as previously described (*Gupta et al., 2010*). In brief, $5 \times 10^4$ cells were plated in the upper chamber in serum-free media. The bottom chamber contained DMEM media with 10% FBS. After 24–48 hr, the bottom of the chamber insert was fixed and stained with Diff-Quick stain. Cells on the stained membrane were counted under a dissecting microscope. Each membrane was divided into four quadrants and an average from all four quadrants was calculated.

## Mice tail vein injection

Mouse tail vein xenografts assay was performed as previously described (*Gupta et al., 2010*). Female athymic nude mice were used. 2.5–3 million iHOT⁺ cells with different Dox treatments in 0.2-ml PBS were injected by the tail vein into individual mice (5–7 for each treatment). Mice were observed generally for signs of illness weekly for the length of the experiment. The lungs were excised and imaged, then fixed in formalin overnight and embedded in paraffin, from which sections were made and stained with hematoxylin and eosin by our pathology consultation service. These slides were examined for the micro-metastases, which were counted in three fields per specimen.

## Bioluminescence imaging

Mice received luciferin (300 mg/kg, 10 min before imaging) and were anesthetized (3% isoflurane) and imaged in an IVIS spectrum imaging system (Xenogen, part of Caliper Life Sciences). Images were analyzed with Living Image software (Xenogen, part of Caliper Life Sciences). Bioluminescent flux (photons $s^{-1}$ $sr^{-1}$ $cm^{-2}$) was determined for the lungs.

## Immunofluorescence

Cells were fixed using 4% paraformaldehyde solution for 20 min at room temperature. The cells were then permeabilized and blocked in blocking solution (0.1% Triton X-100 and 0.1% bovine serum albumin in PBS) for 45 min. Afterward, the cells were incubated with anti-Vimentin (ab92547, abcam) and anti-E-cadherin (14-3249, ebioscience) antibodies at 4°C for 1 hr, followed by Alexa Fluor 546- and 488-labeled secondary antibody for 1 hr, and counterstained with DAPI.

## SmFISH

Dox⁻ and Dox⁺ cells were fixed for 15 min in 4% paraformaldehyde (15714; Electron Microscopy Sciences) in 1× PBS at room temperature, washed three times with 1× PBS, permeabilized for 10 min with 0.5% (vol/vol) Triton X-100 (T8787; Sigma) in 1× PBS at room temperature, and washed three times with 1× PBS. Permeabilized cells were incubated for 5 min in wash buffer comprising 2× SSC (AM9763; Ambion), 30% (vol/vol) formamide (AM9342; Ambion) and 0.1% (vol/vol) murine RNase inhibitor (M0314L; New England Biolabs). Then 500 µl of 1 µM FISH probes (probe design described in *Moffitt et al., 2016*, each probe has an overhang to recruit a fluorophore-conjugated readout probe also described in *Moffitt et al., 2016*) in hybridization buffer was added to cell culture and samples then were incubated at 37°C overnight. Hybridization buffer is composed of wash buffer supplemented with 0.1% (wt/vol) yeast tRNA (15401-011; Life Technologies), 1% (vol/vol) murine RNase inhibitor (M0314L; New England Biolabs), and 10% (wt/vol) dextran sulfate (D8906-50G; Sigma). Cells then were washed with wash buffer and incubated at 47°C for 30 min; this washing step was repeated once. Cells were stained with DAPI during the second washing by adding 10 µg/ml DAPI to the encoding wash buffer. Cells were then washed once with 2× SSC. Fluorophore-conjugated readout probe was further hybridized to FISH probes by incubating cells with 2 nM readout probe in readout hybridization buffer comprising 2× SSC and 10% (vol/vol) ethylene carbonate (E26258; Sigma-Aldrich) for 7 min at room temperature. Cells were then washed once with readout hybridization buffer for 5 min at room temperature. Imaging was performed using Nikon Spinning Disk Confocal Microscope at CSIF Shriram imaging center at Stanford.

## RNA-seq

Total RNA of iHOT⁺ cells with different treatments were extracted by Trizol following RNA clean-5 kit. Poly-A-selected RNA was isolated and the libraties were prepared with the dUTP protocol and sequenced using the Illumina Genome Analyzer IIX platform with 36-bp reads. Raw reads were aligned to the mouse reference sequences NCBI Build 37/mm9 with STAR aligner. Expression levels of RefSeq annotated genes were calculated using RSEM. The differential analysis was conducted using R package DESeq2. The Benjamini–Hochberg procedure was used to adjust for multiple hypothesis testing. Genes with FDR <0.05 and fold change of 2 were considered as significant.

## ATAC-seq

ATAC-seq of iHOT⁺ cells with different treatments was performed as previously described (*Corces et al., 2017*). Briefly, 50,000 cells were pelleted, resuspended in 50 µl of lysis buffer (10 mM Tris–HCl,

pH 7.4, 3 mM MgCl$_2$, 10 mM NaCl, 0.1% NP-40 [Igepal CA-630]), and immediately centrifuged at 500 × g for 10 min at 4°C. The nuclei pellets were resuspended in 50 µl of transposition buffer (25 µl of 2× TD buffer, 22.5 µl of distilled water, 2.5 µl of Illumina Tn5 transposase) and incubated at 37°C for 30 min. Transposed DNA was purified with the MinElute PCR Purification kit (Qiagen), and eluted in 10 µl of EB buffer.

Adaptor sequence trimming using in-house software and mapping to mm9 using Bowtie2 were performed. The reads were then filtered for mitochondrial reads, low-quality reads, PCR duplicates, and black list regions. The filtered reads for each group were merged, and peak calling was performed by MACS2. The reads in peaks for each individual sample were quantified using BEDTools multicov with the merged MACS2 narrow peaks. Peak counts were then combined into an $N \times M$ matrix where $N$ represents called peaks, $M$ represents the samples, and each value $D_{i,j}$ represents the peak intensity for respective peak $i$ in sample $j$. Each peak was annotated by the nearest gene. This matrix was then normalized using R package DESeq2, and differential analysis was conducted using negative binomial models. The Benjamini–Hochberg procedure was used to adjust for multiple hypothesis testing. Peaks with FDR <0.05 and fold change of 2 were considered as significant.

## Acknowledgements

Funding: This project was supported by the National Key R&D Program of China (No: 2022YFA0912900 to QM; 2021YFA1100400, to LL), the National Natural Science Foundation of China (No: 32070870 to QM; 32070867 to LL), Guangdong Basic and Applied Basic Research Foundation (No: 2021 A1515010758 to QM), Guangdong Provincial Key Laboratory of Synthetic Genomics (to QM), Shenzhen Key Laboratory of Synthetic Genomics (No: ZDSYS20180206 1806209 to QM), the Strategic Priority Research Program of the Chinese Academy of Sciences (No: XDPB18 to QM), Program for Oriental Scholars of Shanghai Universities (to LL), Shanghai Frontiers Science Center of Cellular Homeostasis and Human Diseases (to LL), Natural Science Foundation of Shanghai (No: 21ZR1435900 to LL), the NIH grant R35-CA209919 (to HYC) and Howard Hughes Medical Institute (HYC is an HHMI Investigator).

## Additional information

### Competing interests

Howard Y Chang: Reviewing editor, *eLife*. The other authors declare that no competing interests exist.

### Funding

| Funder | Grant reference number | Author |
| --- | --- | --- |
| National Natural Science Foundation of China | 32070870 | Qing Ma |
| National Natural Science Foundation of China | 32070867 | Lingjie Li |
| Guangdong Basic and Applied Basic Research Foundation | 2021A1515010758 | Qing Ma |
| Guangdong Provincial Key Laboratory of Synthetic Genomics | | Qing Ma |
| Shenzhen Key Laboratory of Synthetic Genomics | ZDSYS20180206 1806209 | Qing Ma |
| Chinese Academy of Sciences | Strategic Priority Research Program XDPB18 | Qing Ma |
| Program for Oriental Scholars of Shanghai Universities | | Lingjie Li |

| Funder | Grant reference number | Author |
| --- | --- | --- |
| Shanghai Frontiers Science Center of Cellular Homeostasis and Human Diseases | | Lingjie Li |
| Natural Science Foundation of Shanghai | 21ZR1435900 | Lingjie Li |
| National Key Research and Development Program of China | 2021YFA1100400 | Lingjie Li |
| National Institutes of Health | R35-CA209919 | Howard Y Chang |
| Howard Hughes Medical Institute | | Howard Y Chang |
| National Key Research and Development Program of China | 2022YFA0912900 | Qing Ma |

The funders had no role in study design, data collection, and interpretation, or the decision to submit the work for publication.

## Author contributions

Qing Ma, Conceptualization, Formal analysis, Supervision, Funding acquisition, Investigation, Visualization, Methodology, Project administration, Writing – review and editing; Liuyi Yang, Data curation, Formal analysis, Validation, Visualization, Writing – original draft, Writing – review and editing; Karen Tolentino, Formal analysis, Validation, Investigation, Writing – original draft, Writing – review and editing; Guiping Wang, Data curation, Investigation, Visualization, Helped design and imaging of ASO knockdown of HOTAIR and smFISH for the revision; Yang Zhao, Data curation, Software, Formal analysis, Investigation, Visualization; Ulrike M Litzenburger, Investigation; Quanming Shi, Lin Zhu, Data curation, Investigation, Writing – review and editing; Chen Yang, Data curation, Visualization, Performed the sequencing data analysis for the revision; Huiyuan Jiao, Data curation, Performed the sequencing data analysis for the revision; Feng Zhang, Data curation, Performed the sequencing data analysis for the revision; Rui Li, Investigation, Helped perform the assays for the revision; Miao-Chih Tsai, Howard Y Chang, Conceptualization, Resources, Supervision, Funding acquisition, Investigation, Methodology, Project administration, Writing – review and editing; Jun-An Chen, Data curation, Investigation, Visualization; Ian Lai, Hong Zeng, Data curation, Investigation; Lingjie Li, Conceptualization, Formal analysis, Funding acquisition, Investigation, Visualization, Methodology, Project administration, Writing – review and editing

## Author ORCIDs

Qing Ma (ID) http://orcid.org/0000-0002-8361-5275
Liuyi Yang (ID) http://orcid.org/0000-0001-5405-6411
Jun-An Chen (ID) http://orcid.org/0000-0001-9870-3203
Ian Lai (ID) http://orcid.org/0000-0001-8662-7290
Lingjie Li (ID) http://orcid.org/0000-0003-4348-516X
Howard Y Chang (ID) http://orcid.org/0000-0002-9459-4393

## Ethics

This study was performed in strict accordance with the recommendations in the Guide for the Care and Use of Laboratory Animals of the National Institutes of Health. All mouse work was performed according to IACUC approved protocols at Stanford University (APLAC-14046). All surgery was performed under sodium pentobarbital anesthesia, and every effort was made to minimize suffering.

## Decision letter and Author response

Decision letter https://doi.org/10.7554/eLife.79126.sa1
Author response https://doi.org/10.7554/eLife.79126.sa2

# Additional files

## Supplementary files
• MDAR checklist

## Data availability
Sequencing data have been deposited in GEO under accession codes GSE201581 and GSE201582.

The following datasets were generated:

| Author(s) | Year | Dataset title | Dataset URL | Database and Identifier |
|---|---|---|---|---|
| Ma Q, Tolentino K, Li L, Zhao Y | 2022 | Chromatin accessibility analysis of inducible HOTAIR overexpression mouse breast cancer cells | https://www.ncbi.nlm.nih.gov/geo/query/acc.cgi?acc=GSE201581 | NCBI Gene Expression Omnibus, GSE201581 |
| Ma Q, Tolentino K, Li L, Zhao Y | 2022 | Transcriptome analysis of inducible HOTAIR overexpression mouse breast cancer cells | https://www.ncbi.nlm.nih.gov/geo/query/acc.cgi?acc=GSE201582 | NCBI Gene Expression Omnibus, GSE201582 |

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
