## [Editor Report]

A valuable new mouse model was developed for studying the functional effects of overexpressing *HOTAIR* and the mechanism of action of *HOTAIR*, and used to demonstrate overexpression of *HOTAIR* promoted breast cancer metastasis to the lung. Mechanistically, *HOTAIR* overexpression changed the chromatin from a closed, inactive form to an open, active form, activating specific genes and pathways and causing an altered cellular state that allowed the cells to metastasize; Importantly, the breast cancer cells depended on continuous *HOTAIR* expression, as *HOTAIR* does not trigger an epigenetic memory upon transient induction. The study offers fundamental insights, based on compelling data.

---

## [Decision Letter]

**Decision letter after peer review:**

Thank you for submitting your article "Inducible lncRNA transgenic mice reveal continual role of *HOTAIR* in promoting breast cancer metastasis" for consideration by *eLife*. Your article has been reviewed by 3 peer reviewers, and the evaluation has been overseen by a Reviewing Editor and Erica Golemis as the Senior Editor. The following individuals involved in the review of your submission have agreed to reveal their identity: Manish Charan (Reviewer #1); Kenneth Nephew PhD (Reviewer #3).

Essential revisions:

1. The idea that *HOTAIR* is controlled by epigenetic loss needs to be documented more robustly. Even though the observed phenotypes could be reverted, it is unclear whether the cancer cells, with *HOTAIR* switched off, had conserved any molecular memory of *HOTAIR* overexpression. One criticism is that the study defines epigenetic memory only at the chromatin level or RNA levels. Cells might have a non-genetic memory that would distinguish them from cells that never experienced a high level of *HOTAIR*, without showing any RNA or ATAC seq differences. One way to tackle this issue would be to turn on *HOTAIR* again and define whether they develop migration and mesenchymal phenotypes more quickly than the control cells, which would experience *HOTAIR* overexpression for the first time. In the study, ATAC-seq and RNA-seq do show a return to normal after *HOTAIR* withdrawal, at least for differentially expressed peaks, but the analysis was done using DE genes and peaks identified and selected using no Dox vs Dox. The authors should run differential analyses between +dox vs WD-dox and control vs WD-dox to identify any RNA or ATAC remaining signals that might pinpoint an *HOTAIR*-mediated memory.

2. The *HOTAIR* RNA-mediated effect is not clear. The authors should characterize whether the iHOT+ cells also lose also their phenotypes in absence of the RNA to finalize the characterization of their mice model. For example, iHOT+ ASO-treated cells could be challenged for their migration capacity in comparison to control.

In addition, the reviewers have made a number of recommendations meant to improve the manuscript. Please try to address points on this list, consolidated below from the three reviews.

Recommendations for the authors:

1. Inclusion of H327me ChIP-seq experiments would be very interesting on many levels, including providing greater insight into the human *HOTAIR*-mouse genome chromatin landscape. While these experiments are strongly suggested for inclusion in the manuscript, these experiments could alternatively be included as a future direction in the Discussion section.

2. It would be useful to characterize *HOTAIR* expression in primary tumors and iHOT cancer cells. To allow the community to subsequently use the mouse model and iHOT cells, and eventually to design drugs for therapeutic purposes, the authors should comprehensively characterize the subcellular localizations of *HOTAIR* in situ (in tumors and cells) using smiFISH, and also the chromatin-associated loci of *HOTAIR* in iHOT+ cells using CHIRP. This set of experiments would further validate the mice model and cells.

---

## [Author Response]

Essential revisions:1. The idea that *HOTAIR* is controlled by epigenetic loss needs to be documented more robustly. Even though the observed phenotypes could be reverted, it is unclear whether the cancer cells, with *HOTAIR* switched off, had conserved any molecular memory of *HOTAIR* overexpression. One criticism is that the study defines epigenetic memory only at the chromatin level or RNA levels. Cells might have a non-genetic memory that would distinguish them from cells that never experienced a high level of *HOTAIR*, without showing any RNA or ATAC seq differences. One way to tackle this issue would be to turn on *HOTAIR* again and define whether they develop migration and mesenchymal phenotypes more quickly than the control cells, which would experience *HOTAIR* overexpression for the first time. In the study, ATAC-seq and RNA-seq do show a return to normal after *HOTAIR* withdrawal, at least for differentially expressed peaks, but the analysis was done using DE genes and peaks identified and selected using no Dox vs Dox. The authors should run differential analyses between +dox vs WD-dox and control vs WD-dox to identify any RNA or ATAC remaining signals that might pinpoint an *HOTAIR*-mediated memory.

We thank the reviewer for their helpful suggestions. The reviewer was interested in whether the Dox^WD^ cells develop any molecular memory.

The first way to assay molecular memory is to compare RNA-seq and ATAC-seq profiles of Dox^-^ and Dox^WD^ cells as the reviewer suggested. We compared Dox^-^ vs Dox^WD^ groups and identified very few DEGs (2 up-regulated and 15 down-regulated) by RNA-seq and similarly few differential peaks by ATAC-seq (0 up-regulated and 40 down-regulated) (Figure 4—figure supplement 1C&E). This result suggests that Dox^-^ and Dox^WD^ cells are quite similar in gene expression levels and chromatin status. In addition, these DEGs and differential peaks showed little overlap with DEGs or differential peaks within Dox^+^ and Dox^-^ groups (Figure 4—figure supplement 1D&F), indicating that there were very few remaining signals that might suggest *HOTAIR*-mediated molecular memory. (We have added these results in our manuscript in line 369-372 & line 403-407.)

Second, we determined whether Dox^WD^ cells respond differently compared to Dox^-^ control cells when re-inducing *HOTAIR* expression. We turned on *HOTAIR* in both the Dox^-^ cells and the Dox^WD^ cells by treating with Dox and compared EMT markers (Vimentin and E-cadherin) between both cell lines. After treating with Dox for either 5 or 7 days, we observed no obvious changes in Vim and E-cad levels in both Dox^-^ and Dox^WD^ groups. After treating with Dox for 11 days, E-cad fluorescence intensity decreased in both the Dox^-^ and Dox^WD^ groups, but Vim fluorescence intensity increased only in the Dox^-^ group. Our results suggest that re-inducing high levels of *HOTAIR* in Dox^WD^ cells do not cause Dox^WD^ cells to undergo EMT more quickly than Dox^-^ cells which have never been exposed to *HOTAIR* overexpression, but perhaps even slower. However, upon repeating the experiment several more times, we observed large variations and batch effects between experiments, making it difficult to draw conclusions. Using FISH, we found that *HOTAIR* induction between cell lines was quite heterogenous, as shown in Figure 3—figure supplement 1E in the manuscript. The data suggests that re-inducing *HOTAIR* in Dox^WD^ cell groups may result in a mix of cells which have never experienced *HOTAIR* overexpression or have experienced *HOTAIR* overexpression once or twice. The observed heterogeneity would make the interpretation of *HOTAIR* re-induction rather complicated. We will further pursue whether *HOTAIR* on-off cells show phenotypic memory in our future work.

**Author response image 1. sa2fig1:** 

2. The *HOTAIR* RNA-mediated effect is not clear. The authors should characterize whether the iHOT+ cells also lose also their phenotypes in absence of the RNA to finalize the characterization of their mice model. For example, iHOT+ ASO-treated cells could be challenged for their migration capacity in comparison to control.

Thank you for the helpful suggestions. We have tried several ASOs, but unfortunately the knockdown efficiency of the tested species is not good. The best efficiency we have observed to date is about 44%. Therefore, it is experimentally challenging to test the migration capacity of ASO-treated iHOT+ cells. As we know that most commercially available synthetic modified ASOs are designed to target intronic sequences, which are ideal targets of RNase H in the nucleus and cause splicing defects or RNA degradation more efficiently. However, in our transgenic model, the human *HOTAIR* cDNA sequence inserted into the mouse genome does not contain introns. Therefore, the ASO-mediated knockdown approach may not be the best strategy for this model.

In addition, the reviewers have made a number of recommendations meant to improve the manuscript. Please try to address points on this list, consolidated below from the three reviews.Recommendations for the authors:1. Inclusion of H327me ChIP-seq experiments would be very interesting on many levels, including providing greater insight into the human *HOTAIR*-mouse genome chromatin landscape. While these experiments are strongly suggested for inclusion in the manuscript, these experiments could alternatively be included as a future direction in the Discussion section.

Thanks for the helpful comments. We have included this suggestion as a future direction in the Discussion section (line 624-628).

2. It would be useful to characterize *HOTAIR* expression in primary tumors and iHOT cancer cells. To allow the community to subsequently use the mouse model and iHOT cells, and eventually to design drugs for therapeutic purposes, the authors should comprehensively characterize the subcellular localizations of *HOTAIR* in situ (in tumors and cells) using smiFISH, and also the chromatin-associated loci of *HOTAIR* in iHOT+ cells using CHIRP. This set of experiments would further validate the mice model and cells.

Thank you for your suggestion! We have analyzed the *HOTAIR* expression levels in primary tumors using qRT-PCR as shown in Figure 2B.

For iHOT cells isolated from the tumors, we analyzed the subcellular localizations of *HOTAIR* in iHOT cells using smFISH according to the reviewer’s suggestions. As shown in Figure 3—figure supplement 1E, our results reveal that *HOTAIR* localizes to both the nucleus and cytoplasm. The induction of *HOTAIR* is quite heterogeneous across cell populations. (We have added this result in the manuscript in line 262-265.)

We also further analyzed the published *HOTAIR* ChIRP-Seq data to see whether *HOTAIR*-associated genes are altered with *HOTAIR* induction in iHOT cells. The previous study identified 832 *HOTAIR* occupancy sites genome-wide in human breast cancer cells, which are associated with 1345 genes (*HOTAIR* targets) in total (Chu et al., *MolCell*, 2011). We compared *HOTAIR* targets established from previous literature with DEGs and differential peak-associated genes identified from our RNA-seq and ATAC-seq assays on Dox^-^ versus Dox^+^ iHOT cells. As shown in Figure 4—figure supplement 2A, we observed significant overlaps between *HOTAIR* targets and ATAC-seq differential peak-associated genes (207 genes, ~13% of ATAC-seq differential peak-associated genes). Among these shared genes, the chromatin accessibility of 116 genes were up-regulated and 91 were down-regulated. Our gene ontology (GO) analysis uncovered terms such as cell adhesion, EMT or migration (Figure 4—figure supplement 2A). We also noted significant overlap between *HOTAIR* targets and DEGs in our RNA-seq data (34 total, 13 genes were up-regulated and 21 genes were down-regulated in Dox^+^ cells compared to control, Figure 4—figure supplement 2B). GO analysis of these genes also revealed associations with multiple EMT and migration related terms (Figure 4—figure supplement 2B). Furthermore, we compared the *HOTAIR* targets to genes regulated by *HOTAIR* through chromatin accessibility, which were represented as overlapping genes shared between our ATAC-seq and RNA-seq datasets (Figure 4E). We identified 12 genes out of a total of 92 genes (~13%) which may be direct targets regulated by *HOTAIR* in iHOT cells (Figure 4—figure supplement 2C). Interestingly, almost all of these genes were associated with migration. Genes shown to promote migration such as *Ecm1*, *Gpr39*, *Padi1* were up-regulated and genes shown to suppress migration such as *Limch1* were down-regulated (Figure 4—figure supplement 2C). These results suggest that *HOTAIR* regulates the expression of multiple cell adhesion and EMT associated genes by regulating their chromatin accessibility to promote tumor metastasis. (We have added these results to the Discussion section in line 596-623.)

­­